# Sarcoidosis, *Mycobacterium paratuberculosis* and Noncaseating Granulomas: Who Moved My Cheese

**DOI:** 10.3390/microorganisms11040829

**Published:** 2023-03-24

**Authors:** Coad Thomas Dow, Nancy W. Lin, Edward D. Chan

**Affiliations:** 1McPherson Eye Research Institute, University of Wisconsin, Madison, WI 53705, USA; 2Division of Environmental and Occupational Health Sciences, National Jewish Health, Denver, CO 80206, USA; 3Division of Pulmonary Sciences and Critical Care Medicine, University of Colorado School of Medicine, Aurora, CO 80045, USA; 4Department of Academic Affairs, National Jewish Health, Denver, CO 80206, USA; 5Rocky Mountain Regional Veterans Affairs Medical Center, Department of Medicine, Aurora, CO 80045, USA

**Keywords:** sarcoidosis, *Mycobacterium avium* subsp. *paratuberculosis*, MAP, non-caseating granuloma, caseating granuloma, Blau syndrome, early onset sarcoidosis, Crohn’s disease, cell wall deficient (CWD), L-form, *Mycobacterium tuberculosis*, acid-fast bacilli, *SLC11a1* (*NRAMP1*), *NOD2* (*CARD15*)

## Abstract

Clinical and histological similarities between sarcoidosis and tuberculosis have driven repeated investigations looking for a mycobacterial cause of sarcoidosis. Over 50 years ago, “anonymous mycobacteria” were suggested to have a role in the etiology of sarcoidosis. Both tuberculosis and sarcoidosis have a predilection for lung involvement, though each can be found in any area of the body. A key histopathologic feature of both sarcoidosis and tuberculosis is the granuloma—while the tuberculous caseating granuloma has an area of caseous necrosis with a cheesy consistency; the non-caseating granuloma of sarcoidosis does not have this feature. This article reviews and reiterates the complicity of the infectious agent, *Mycobacterium avium* subsp. *paratuberculosis* (MAP) as a cause of sarcoidosis. MAP is involved in a parallel story as the putative cause of Crohn’s disease, another disease featuring noncaseating granulomas. MAP is a zoonotic agent infecting ruminant animals and is found in dairy products and in environmental contamination of water and air. Despite increasing evidence tying MAP to several human diseases, there is a continued resistance to embracing its pleiotropic roles. “Who Moved My Cheese” is a simple yet powerful book that explores the ways in which individuals react to change. Extending the metaphor, the “non-cheesy” granuloma of sarcoidosis actually contains the difficult-to-detect “cheese”, MAP; MAP did not move, it was there all along.

## 1. Introduction

The first clinical record of sarcoidosis dates to 1859 when Jonathan Hutchinson of the Blackfriers Hospital for Diseases of the Skin of London described a patient with raised, purple skin plaques; initially Hutchinson related them to the patient’s gout. Upon seeing another individual, a Mrs. Mortimer, with similar dermatologic findings, Hutchinson differentiated the lesions from tuberculosis and lupus and labeled the condition “Mortimer’s malady” [1]. The label sarcoidosis came in 1899 by Caesar Boeck with the description “multiple benign sarcoid of the skin” as the histology resembled sarcoma. Boeck also noted the remarkable clinical and histologic similarity to cutaneous tuberculosis and was the first to propose possible link between sarcoidosis and tuberculosis [2]. Pulmonary involvement was not identified in the early discussion of sarcoidosis as this era predated the clinical use of the X-ray [3]. This diagnostic uncertainty of the association of sarcoidosis and tuberculosis can be seen in the title of a small series from 1957: “Stenosing non-caseating tuberculosis (sarcoidosis) of the bronchi” [4]. 

Common to many inflammatory/autoimmune diseases is the concept of an “environmental trigger.” Infectious agents have been the suspected cause of sarcoidosis since the early 1900’s with mycobacteria considered the likely agent [5,6,7]. In a 1964 series, the sera of nearly 80% of 280 sarcoidosis patients revealed significant reaction to mycobacterial antigens [8]. In a 1968 case report of sarcoidosis, a non-tuberculous mycobacterium (NTM) was grown from the lymph gland of an individual with sarcoidosis—it was labeled “anonymous mycobacteria” [9]. A 1996 study demonstrated the isolation and/or identification of *Mycobacterium avium* subsp. *paratuberculosis* (MAP), or a closely related *M. avium* complex [MAC] strain, from sarcoid skin lesions and cerebrospinal fluid [10].

A hallmark of sarcoidosis is the non-caseating granuloma [11]. However, this is by no means pathognomonic; indeed, a mnemonic developed for the differential diagnosis of non-caseating granulomas used every letter of the alphabet [12].

Caseous necrosis is associated with tuberculosis; *M. tuberculosis* produces a virulence factor, trehalose 6,6′ dimycolate, that is markedly toxic [13]; it is the proposed trigger of caseous necrosis [14]. While immunity limits the number of tuberculous bacilli in immunocompetent individuals (and animals), it does not eliminate them all. The resultant pathology comes, not from increasing mycobacterial burden, but rather the hypersensitivity to the *M. tuberculosis* antigens [15]. However, it is important to recognize that tuberculosis may also be associated with non-caseating granulomas [16] (Chan E.D., unpublished data).

Caseating and non-caseating granulomas are types of inflammatory lesions. Caseating granulomas are characterized by the presence of dead or necrotic tissue that has a *cheesy* consistency—“caseous necrosis”—where histological features are not preserved. Non-caseating granulomas do not contain necrotic tissue and are composed of activated immune cells such as macrophages and T-lymphocytes. Both types of granulomas are found in a variety of conditions; however, caseating granulomas are more commonly associated with bacterial infections such as tuberculosis and non-tuberculous mycobacterial infections, whereas non-caseating granulomas are more commonly associated with “immune-mediated disorders” such as sarcoidosis (Figure 1). Another non-caseating granulomatous disease is Crohn’s disease. Because the ever-increasing association of MAP with Crohn’s disease, paralleling this association is an integral part of this manuscript featuring mycobacterial contribution to non-caseating granulomatous disease [17,18]. Naser made major contributions to the MAP/Crohn’s association with reports culturing MAP from the blood of Crohn’s patients [19] and by culturing MAP from the breast milk of Crohn’s patients [20]. Throughout the 1980’s, Chiodini presented case reports finding MAP in Crohn’s patients [21]. Chiodini brought his MAP experience to bear on sarcoidosis in a 1993 case series where he tested sera from sarcoid patients for reactivity against MAP antigens [22]. The 28 patients with sarcoidosis had a significantly higher IgG values compared to controls (*p* = 0.0009) suggesting a no-longer-anonymous mycobacterium in sarcoidosis but identifying a “mycobacterium of interest” as MAP [22]. 

## 2. *Mycobacterium avium* subsp. *paratuberculosis* (MAP)

MAP belongs to the MAC category that comprises over 10 species and subspecies, some of which are pathogenic to animals and humans [24]. MAP is the cause of Johne’s disease, or paratuberculosis, a fatal enteric infectious disease manifesting as chronic diarrhea and wasting found mostly in ruminant animals [25]. Infected animals shed MAP in their milk and feces, a state commonly persisting for years prior to the animal displaying clinical Johne’s disease. Johne’s disease is common; the United States Department of Agriculture (USDA) reported that the herd-level prevalence of MAP infection in U.S. dairy herds increased from 21.6% in 1996 to 91.1% in 2007 [26].

Inefficiency of diagnostic testing for MAP along with the long latency of infection in apparent uninfected yet productive dairy animals create hesitancy by producers to test their herds; moreover, there is no mandate requiring testing. Collectively, this results in more MAP-shedding animals that propagate infection and contaminate milk and meat, promotes trade of asymptomatic animals, and delays the mainstay by which Johne’s disease is controlled, animal culling [27].

MAP is a resilient organism and shedding by infected animals is a major source of environmental MAP; once excreted, MAP can survive up to 120 weeks in soil or water [27]. MAP is found in grazing areas as well as in runoff continuing to rivers and in municipal water [28].

Both Iceland and what is now the Czech Republic were historically isolated from Johne’s disease due to limited importing/exporting of ruminant animals. For Iceland this changed in 1933 when apparently healthy, but sub-clinically MAP-infected sheep were imported [29]. Other sheep and cattle on the same importing farms acquired Johne’s disease with the same MAP strain [30]. Similarly, until 1989 Czechoslovakia was isolated; open borders and importation of livestock followed the lifting of the Iron Curtain. Crohn’s disease in both the Icelandic and Czech populations became a lagging indicator of MAP infection [31].

## 3. MAP Is Difficult to Detect

Regarding human Crohn’s samples, the basis of the hundred-year controversy is the fact that traditional culturing and staining are largely unsuccessful in identifying MAP [18,32,33]. Aside from *M. leprae*, the microbial cause of leprosy that has never been grown in culture, MAP is the slowest growing of the mycobacteria with some labs continuing for a year before calling a culture MAP-negative. Veterinary laboratories employing both enhanced conventional and molecular methods for identifying MAP in ruminant animals such as cattle and sheep have the greatest experience in detecting MAP [34]. 

In addition to being difficult to culture, MAP can exist with a modified cell wall—the component of the mycobacterium that takes up a characteristic acid-fast stain. MAP can shed its cell wall, becoming a spheroplast or L-form [35]. The bacterium then cannot be detected microscopically in the traditional manner. This morphologic change allows some MAP to become spore-like. The spore morphotype capable of surviving heat and other stressors enables MAP to persist in host macrophages and in the environment [36].

Humans are exposed to MAP in contaminated food, water, and air [37]. A primary reason that MAP is such a risk for human consumers is because this bacterium is particularly resistant to heat [38]. Milk and dairy products are the primary sources of MAP infection in humans [39], and pasteurization only reduces the MAP load in milk [39,40]. MAP is present in yogurt [41], muscle meat [42], and hamburgers [43]. Alarmingly, viable MAP is found in infant formula, including infant formula powder [44,45,46].

For this review, MAP in cheese is featured. The U.S. Food and Drug Administration allows two options for producing safe cheese products: either use pasteurized milk in cheese production or cure the finished cheese for at least 60 days at temperatures of 2 °C. The curing process can include low pH and various salt concentrations to lessen pathogens; like pasteurization, efforts in the curing process to curtail MAP have some, but not total success, with MAP in cheeses [47]. MAP is found in Swiss cheese [48] and persists in the manufacture and ripening of cheddar cheese [49]. Cheese from sheep and goat as well as combinations of these contain MAP [50]. Likewise, MAP is detected in the pasta-filata cheeses such as mozzarella, burrata, and provolone [51].

To date, MAP has been associated with Crohn’s disease, Blau syndrome, autoimmune (Hashimoto’s) thyroiditis, autoimmune diabetes, multiple sclerosis, lupus erythematosus, rheumatoid arthritis, and Parkinson’s disease [52,53]. While it is intuitive to assign a role to MAP in granulomatous disease, MAP’s contribution to inciting autoimmune disease is less obvious; it is envisaged to be due to molecular mimicry to one or more of its proteins, primarily epitope homology to its heat shock protein 65 (HSP65) [54]. 

Machine learning as applied to the microbiome is a nascent science for developing biomarkers of specific disease states. It has recently been applied in cattle to generate a microbiota signature for MAP infection [55]. In a specialized gastrointestinal referral hospital located in Sudan, of 67 participants with microbiome screening, 27 (40%) were MAP positive; moreover, this cohort had a unique microbiome profile compared to the MAP-negative participants [56]. The authors note that the results may indicate that a considerable proportion of the Sudanese population could be MAP infected or carriers and this may be because in Sudan, there is frequent close contact between humans and livestock. The pulmonary microbiome is increasingly recognized; results determined by analysis of bronchoalveolar lavage fluid help discriminate between tuberculosis, infections other than tuberculosis, and lung cancer [57]. A single series of 35 pulmonary sarcoidosis patients has been reported showing a dysbiosis of the lower airway microbiota [58]. These distinctive probings of enteric and pulmonary microbiomes will likely offer solid clues to direct future investigations on the cause(s) of sarcoidosis.

## 4. Sarcoid Epidemiology

Reports on the epidemiology of sarcoidosis are quite varied. Such disparity is likely due to a number of factors including [59,60]: (i) retrospective analyses using large health administrative data with inherent inaccuracies in diagnostic coding; (ii) bias collection of case series which may not include milder cases; (iii) potential inclusion of mimics of sarcoidosis as sarcoidosis is best considered a syndrome (“sarcoidoses”); that is., sarcoidosis is often considered to be due to an environmental exposure (along with host susceptibility) to various agents and thus different regions of the world are likely to have different inherent differences in incidence and prevalence; and (iv) greater diagnostic detection of (asymptomatic) sarcoidosis with greater use of CT scans for other reasons in resource-rich countries and less so in resource-poor countries. Nevertheless, we have attempted to summarize below the epidemiology of sarcoidosis.

Erdal and co-workers [61] analyzed the electronic medical records of a large institution network (The Ohio State University Medical Center’s Institutional Information Warehouse) from 1995 to 2010 to determine the temporally prevalence of sarcoidosis. Using the ICD9 code for sarcoidosis (ICD9 135), they found that the prevalence increased steadily from 164/10^5^ in 1995 to 330/10^5^ in 2010. When they examined the prevalence of sarcoidosis in Columbus, Ohio (Franklin County)—a region with a demographic profile (race, gender, age) that is nearly identical to the U.S. as a whole—the prevalence of sarcoidosis was estimated to be 48/10^5^ in 2010 (and may be as high as 200/10^5^), significantly higher than the U.S. The authors speculated that although an actual increase in the number of sarcoidosis cases could be occurring over this 15-year period, greater use of CT scans, the availability of less invasive diagnostic techniques, and perhaps increased awareness of sarcoidosis by clinicians likely played a role in the increased prevalence.

In a similar study from Korea of sarcoidosis between 2007 and 2016—coded by the Korean Classification of Disease—the annual incidence rates of sarcoidosis increased from 0.85/10^5^ in 2009 to 0.97/10^5^ in 2015 [62]. The highest incidence rate occurred in those 50–59 years for females, while biphasic peaks were seen in males (30–39 years and 60–69 years). In this population that is racially different and more homogenous than the U.S. population, the 10-year prevalence was estimated to be 11.44/10^5^ in females and 7.30/10^5^ in males (combined = 9.37/10^5^).

In a study from Ontario, Canada, the prevalence of sarcoidosis increased from 66/10^5^ in 1996 to 143/10^5^ in 2015, a highly significant increase of 116% [63]. Interestingly, the incidence of sarcoidosis decreased from 7.9/10^5^ in 1996 to 6.8/10^5^ in 2014; when the incidence was stratified by gender, there was a 30% decrease in the incidence in females, but a 5.5% increase in males. The authors speculated that the higher sarcoidosis incidence in males in the latter years examined was due to an increase in occupational dust exposures as there was a significant increase in the construction industry in Ontario during this period.

Seedahmed and co-workers [64] analyzed the Veterans Health Administration electronic health record system to define the epidemiology of sarcoidosis from 2003 to 2019. Among 13 million veterans, there were 23,747 (0.20%) incident diagnosis of sarcoidosis, approximately four-fold greater than the general U.S. population. The annual incidence of sarcoidosis in the veteran population increased from 38 to 52 cases/10^5^ person-years and the annual prevalence from 79 to 141 cases/10^5^ persons [64]. They also found that living in the Northeast and serving in Army, Air Force, and multiple branches were more associated with a diagnosis of sarcoidosis than working in the Navy.

In summary, while numerous studies report increasing prevalence of sarcoidosis in specific countries, it is not known whether this increased prevalence is due to increasing new incident cases (resulting in a true increase in prevalence), greater detection of asymptomatic and symptomatic sarcoidosis with use of CT scans, reporting bias of positive epidemiologic data, inadvertent bias due to selective inclusion of specific racial groups in whom sarcoidosis appear to be more common, or combinations of any of these elements. Even within a country, there are reports of geographic variation of sarcoidosis likely related, in large part, to differences in ethnic group populations studied [65,66,67]. Thus, analyzing the incidence and prevalence of sarcoidosis over time within a country is challenging and fraught with intentional and unintentional biases; comparing the incidence and prevalence of sarcoidosis between countries and assessing the temporal global frequency of sarcoidosis accurately is even more difficult due to greater variations of the aforementioned factors between countries.

## 5. Genetic Risk for Sarcoidosis Overlaps with Risk for Mycobacterial Infection

Sarcoidosis is most likely the result of some form of genetic susceptibility plus an inciting environmental factor. There is a 3.7-fold increase in sarcoidosis with having at least one first degree relative with sarcoidosis [68]. While sarcoidosis may affect all races, the Black race appears to have increased vulnerability. A large U.S. military veteran study found that, using multivariate analysis, Black race, female sex, and history of tobacco use were significantly associated with a diagnosis of sarcoidosis [64]. In general, Black individuals experience more severe disease, multi-organ involvement, and higher rate of hospitalization [60]. Additionally, Black females are reported to have higher sarcoidosis mortality rates [69].

## 6. SLC11a1 (NRAMP) and Sarcoidosis

Risk of sarcoidosis is associated with polymorphisms of the *SLC11A1* gene [70]. What was initially termed natural resistance-associated macrophage protein 1 (NRAMP1) is now referred to as SLC11A1 (solute carrier 11a1). The gene that encodes for this protein is recognized as having a role in the susceptibility of both humans and animals to infections, including mycobacterial infections; moreover, it is associated with several inflammatory and autoimmune diseases. In human beings, the *SLC11A1* gene is located on chromosome 2q35. It encodes an integral membrane protein of 550 amino acids that is expressed exclusively in the lysosomal compartment of monocytes and macrophages [71]. The product of the *SLC11A1* gene modulates the cellular environment in response to activation by intracellular pathogens by acidifying the phagosome thus killing the pathogen [72]. As such, it plays a role in host innate immunity [73]. Mutation of *SLC11A1* impairs phagosome acidification yielding a permissive environment for the persistence of intracellular bacteria [74].

Susceptibility to mycobacterial diseases such as tuberculosis, leprosy, and Buruli’s ulcer, are associated with polymorphism of the *SLC11A1* gene [75,76,77,78]. Similar polymorphisms are associated with Johne’s disease in cattle [79], goats [80], and sheep [81]. When researchers at the Belgium Pasteur Institute developed a murine model for MAP infection, they created an *SLC11A1* defect mouse [82].

In addition to *SLC11A1* and sarcoidosis, polymorphisms of this gene are also found in rheumatoid arthritis [83], inflammatory bowel disease [84,85], multiple sclerosis [86], and autoimmune diabetes [87]. Each of these diseases is also associated with MAP [31,53].

## 7. HLA Alleles and Sarcoidosis/Tuberculosis Risk

The human leukocyte antigen (HLA) system coordinates the adaptive immune response to foreign antigens, which includes both infectious and non-infectious agents. The HLA genes encode major histocompatibility complex (MHC) molecules, which are found on antigen presenting cells that process and present antigens in the context of MHC molecules to T-lymphocytes, resulting in the initiation of the adaptive immune response to that antigen [88]. 

The T-lymphocytes that proliferate in sarcoidosis often show a restricted repertoire of T-cell receptors suggesting a limited number of peptides are responsible for the expansion of these T-lymphocytes [89]. Research in sarcoidosis suggests that these peptides may have mycobacterial origins; for example, mycobacteria can be found in the 23–30% of tissues samples from those with sarcoidosis [90]. 

Many HLA alleles are thought to be associated with sarcoidosis. The U.S. ACCESS study found that HLADRB1 * 1101, DRB1 * 1201, DRB1 * 1501, and DRB1 * 0402 alleles were strongly associated with sarcoidosis [91]. Additionally, HLA-DRB1 * 03:01 was associated with an acute form of sarcoidosis called Lofgren’s syndrome, whereas HLA-DRB1 * 1101 was associated with chronic sarcoidosis [92,93]. A large genome-wide association study of ocular sarcoidosis found a link with HLA-DRB1 * 04:01 and MAGI1 [94]. These findings underscore the importance of analyzing and sarcoid phenotypes separately in genetic studies.

HLA alleles are associated with different risks for tuberculosis as well. HLA-DRB1 * 04 and HLA-DRB1 * 08 genotypes seem to increase the risk for infection, whereas HLA-DRB1 * 03 and HLA-DRB1 * 07 genotypes seem to decrease the risk [88]. It should be noted that genotype distribution is also highly affected by ethnic differences, which makes associations difficult to generalize across populations. 

Shared HLA alleles in individuals with sarcoidosis and *Mycobacterium tuberculosis* infections suggest a possibly common pathogenic pathway between sarcoidosis and mycobacterial infections. As mentioned before, HLA-DRB1 * 1101 has been associated with susceptibility to sarcoidosis. Oswald-Richter et al. demonstrated that individuals with HLA-DRB1 * 1101 were statistically more likely to recognize ESAT-6, a peptide derived from mycobacteria [95]. These authors also showed that antigen presenting cells expressing HLA-DRB1 * 1101 interact with mycobacterial peptides, ESAT-6 and katG, to elicit a T_H_1 polarized response from sarcoidosis T-cells [95]. Interestingly, a dysregulated T_H_1 response is theorized to be the immune dysfunction responsible for sarcoidosis. Another study, using a bioinformatic approach, demonstrated that individuals with sarcoidosis tended to express HLA-DR allele combinations that recognize *M. tuberculosis* and *M. avium* epitopes with greater affinity than those individuals with tuberculosis, suggesting that sarcoidosis may be a hypersensitive reaction to mycobacterial antigens [90]. These results suggest that HLA genes and mycobacterial peptides may contribute to the immune dysregulation seen in sarcoidosis and may provide a mechanistic explanation for the etiology of sarcoidosis.

HLA polymorphisms in individuals with sarcoidosis and *M. tuberculosis* infections may contribute to different sub-phenotypes in both diseases. In a study by Dawkin et al., it was found that certain HLA alleles were found in higher frequencies in individuals who never developed *M. tuberculosis* infections despite close contact with an index case and in those with acute/resolved sarcoidosis compared to individuals with *M. tuberculosis* infections and chronic sarcoidosis, respectively [96].

The different phenotypes of sarcoidosis may be the result of how different HLA alleles respond to mycobacterial peptides. Grosser *et al.* found that using HLA-DRB1 alleles in combination with *M. tuberculosis* status enabled the authors to predict the sarcoidosis phenotype: sarcoidosis cases with either HLA-DRB1 * 03 or -DRB1 * 04 allele and the absence of *M. tuberculosis* DNA were associated with acute/resolved disease, while cases with HLA-DRB1 * 11, -DRB1 * 15, and the presence of *M. tuberculosis* DNA were associated with chronic sarcoidosis [97]. The authors hypothesized that *M. tuberculosis* could be an antigen that produces a more severe clinical course in sarcoidosis with certain HLA alleles [97]. The study by Saltini et al. supports this hypothesis, showing that individuals with acute sarcoidosis had HLA-DR genomes with an exaggerated ability to recognized *M. tuberculosis* and *M. avium* epitopes, whereas those with chronic versions of the disease had HLA alleles with lower affinity to the same epitopes [90]. The authors posited this could represent an efficient immune response to mycobacterial antigens in acute sarcoidosis with subsequent clearance and resolution of disease. The HLA alleles associated with chronic sarcoidosis may produce a less efficient response to mycobacterial antigens—due in part to a lower affinity for these epitopes—resulting in antigen persistence, continuous granuloma formation, and chronic disease [90]. 

## 8. CARD15 (NOD2)—Early Onset Sarcoidosis (EOS) and Blau Syndrome

The *CARD15* gene is part of the ancestral innate immune system that encodes for a cytoplasmic receptor that senses and eliminates bacteria [98]; originally referred to as the *NOD2*, it is now known as the *CARD15* gene [99]. Blau syndrome is a rare disease with the clinical triad of dermatitis, arthritis, and uveitis. It is the only inflammatory disease that is autosomal dominantly inherited and is monogenic. Blau syndrome has generated interest in medical literature because of the discovery that places its genetic defect with the same Crohn’s susceptibility *CARD15* gene [100,101]. 

The component of *CARD15* associated with Blau syndrome susceptibility is found within the nucleotide binding site domain [102], whereas the Crohn’s susceptibility is in the N-terminal leucine- rich repeat domain [103]. In a small series looking for MAP in six individual Blau tissues, including skin, synovium, and granulomas of the liver and kidney, all six tissues had the DNA of MAP [104]. Early onset sarcoidosis (EOS) is the identical clinical presentation as Blau syndrome but without the family history [105,106].

## 9. An Illustrative Case of Cardiac Sarcoidosis

Cardiac involvement is an increasingly recognized and serious manifestation of sarcoidosis ranging from silent myocardial granulomas leading to sudden death [107,108,109] to symptomatic arrhythmias and heart failure [110]. In isolation, cardiac sarcoidosis can be very difficult to detect [111]. In a patient with sarcoidosis who died of a heart attack, MAP was detected in the noncaseating granuloma that infiltrated the cardiac tissue [112].

A particularly instructional sarcoidosis case was reported in 2018: *Case Study: Cardiac sarcoidosis resolved with Mycobacterium avium* subsp. *paratuberculosis antibiotics (MAP)* [113]. The patient, a physician, had inactive sarcoidosis later followed by acute cardiac sarcoidosis that manifested as complete heart block, cardiomyopathy, and heart failure. His own literature search led him to the prospect of MAP as a cause for sarcoidosis. He sent blood to the only commercial lab culturing human samples for MAP, Otakaro Pathways in New Zealand (https://otakaropathways.co.nz (accessed on 3 March 2023)). His blood was positive for MAP. The MAP-targeting anti-mycobacterial therapy [114] resulted in complete resolution of his cardiac sarcoidosis and no residual pulmonary nodules nor mediastinal or distant lymphadenopathy [113]. This parallels similar MAP-targeting antibiotic therapy for Crohn’s disease [115,116,117].

Although a mainstay of sarcoidosis treatment, corticosteroids are palliative but not curative with potentially major side effects [118]. Up to 74% relapse after stopping corticosteroids [119]. Individuals with immune-mediated diseases such as rheumatoid arthritis have a higher probability to develop active tuberculosis with use of corticosteroids compared to the general population [120]. This highlights the immune component of the disease in an individual’s management of exposure to tuberculosis. While latent tuberculosis infection indicates the ability to contain mycobacteria by the host immune response, active TB is an indication of a failure of the immune response to contain *M. tuberculosis* [121]. Variation on this theme may extend to sarcoidosis and Crohn’s disease as well. Antibiotic therapy targeted against MAP in Crohn’s disease showed that such treatment can play a significant role in inducing a protective immune response in macrophages, endothelial cells, and T lymphocytes, even in absence of MAP infection [122]. Likewise, studies have shown antimycobacterial therapy decreases the level of mycobacterial antigens in pulmonary sarcoidosis, but more studies are needed to determine the true effectiveness of therapy on disease course [123]. 

## 10. Discussion

An entrenched tenant is that sarcoidosis is a disease of undetermined etiology. While the parallel non-granulomatous story of MAP and Crohn’s disease has garnered an increasingly tighter association, a MAP/sarcoidosis association remains to be made known; this article proposes a causal role for MAP in sarcoidosis. As presented in various sections of this paper, MAP has been found in juvenile sarcoidosis tissues and anti-MAP antibiotic therapy has successfully treated cardiac sarcoidosis. Revealed in the epidemiology section of this article, sarcoidosis is increasing in its prevalence; also, the genetics section of this article indicates that the risks for sarcoidosis are shared risks for mycobacterial infection. Parsimony favors a causal role for mycobacteria in sarcoidosis. A mycobacterium to which humans are regularly exposed is MAP, a mycobacterium that is difficult to detect. A consensus article calling for MAP zoonotic designation was the product of a 2017 international assembly of MAP researchers—such precautionary measures have yet to be taken [124].

The book “Who Moved My Cheese?” by Spencer Johnson, M.D. is about four characters who live in a maze and rely on finding cheese to survive [125]. The cheese in the story represents what people seek and the way that different individuals respond to change and uncertainty in their lives. The book encourages readers to be adaptable and flexible in the face of change, and to embrace new opportunities even when they may be uncomfortable or unfamiliar.

This article, like the book, asks the reader to consider something unfamiliar: the inertia of MAP’s causal involvement in an increasing list of inflammatory and autoimmune diseases as it applies to sarcoidosis [53]. Understanding the limitations of conventional laboratory investigations as they relate to detecting MAP, molecular investigation employed by dedicated veterinary laboratories are the key to exposing MAP as the previous anonymous mycobacterium associated with sarcoidosis.

In the book, characters searching the maze for cheese leave messages on the wall—they do this in hopes that those who follow will “see the writing on the wall.” Extending the metaphor, the writing on the wall of this article is that although there is no apparent cheesy necrosis in the noncaseous granulomas of sarcoidosis, the sought-after cheese (“MAP”) was there all along.

## Figures and Tables

**Figure 1 microorganisms-11-00829-f001:**
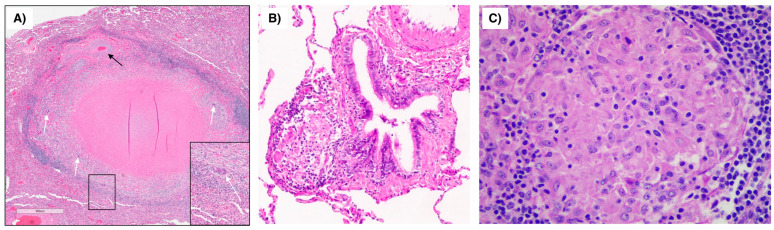
**Pulmonary granuloma due to mycobacteria vs. sarcoidosis.** (**A**) Granuloma with central necrosis surrounded by epithelioid histiocytes, multinucleated giant cells (white arrows) and fibrosis in the resected left lower lobe of a patient with *Mycobacterium avium* complex lung disease. Note the adjacent pulmonary artery (thin black arrow). Inset shows a magnified multinucleated giant cell. (**A**) is modified from Schenkel AR et al. [23] *Immune Netw* 2022. (**B**) Sarcoid granuloma (arrow) next to a medium size ciliated bronchus. (**C**) Magnified view of another sarcoid granuloma. (**B**,**C**) are courtesy of Carlyne Cool, MD, National Jewish Health.

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
