# Peer review of "Sarcoidosis, Mycobacterium paratuberculosis and Noncaseating Granulomas: Who Moved My Cheese"

_microorganisms, 2023, doi:10.3390/microorganisms11040829_

Round 1
Reviewer 1 Report
Dear Colleagues,
I had the privilege of reviewing this paper and came up with the following points and suggestions that maybe of interest:
Can NTM cause pleomorphic disease – if so; how and why?
Possible inclusion of how does MAP cause disease after infection? Why would only a sub-population develop disease? Is it in humans like in ruminants, increased gut epithelial permeability? How would it do this in the lung? This is not essential to be included but worth a thought.
Distinction of caseating vs non-caseating – pathological processes and why it may come about – relates to the pathogenicity of the mycobacterial organism?
One reluctance to the acceptance of infectious causation of sarcoid and Crohn's - common doctrine that the use of steroids and immunosuppressives would reactivate NTM like it does with TB, and worsen clinical condition (although steroids used in cerebral TB). Highlight differences between these groups also in their response to therapy – inc resistance to traditional Anti-TB drugs. Immune containment.
Mention studies of treatment of sarcoidosis with AMAT are required to assess effectiveness.
Page 4 paragraph 2
It is not clear that the CWDM are all in spore form but more acceptable to say ‘This morphologic change allows some MAP to become spore-like’.
From the referenced paper
“MAP spores display enhanced infectivity as well as maintain acid-fast characteristics (please check paragraph in paper which is different) upon germination in a well-established bovine macrophage model. This is the first study to demonstrate a new MAP morphotype possessing spore-like qualities”
Same page, last paragraph
Can also mention the involvement of the gut microbiome as the originator and co-conspirator in development of various autoimmune disease implicated by MAP. Recognise and acknowledge limitations that those illnesses do have or have not granulomas in their pathology. Also the possible role of the lung microbiome that is being seen to be abnormal in TB.
Summary of epidemiology section. Maps of countries do help but otherwise - is sarcoid growing exponentially across the world and in previously low incidence countries?
One paper has mentioned: The prevalence of sarcoidosis was 143 per 100 000 in 2015, increasing by 116% (p<0.0001) from 1996.
Page 8
Loss of paragraph continuity after 2nd line – maybe format of paper downloaded.
Kind Regards,
Author Response
Thank you for your constructive review. Our reply is in the attached Word document.

Reviewer 2 Report
The manuscript entitled "Sarcoidosis, Mycobacterium paratuberculosis and Noncaseating Granulomas: Who Moved My Cheese" is an interesting paper on MAP and sarcoidosis. It was written and organized well. I have some recommendations to improve it:
- Please always write the Latin species names italic. However, it should not be italic in an italic text (e.g. page 9 first paragraph).
- Page 3: "complex" in "Mycobacterium avium complex" should not be italic.
- Please use "subsp." for subspecies, with a dot at the end and not as "ss".
- Abbreviations: First letter of "paratuberculosis" should be small.
Author Response
Thank you for the constructive review of our manuscript. Your suggestions were primarily related to formatting and we have incorporated all of them into the updated manuscript.